# Salience and perceptions of epidemic-prone diseases in two communities: Findings from freelisting interviews in Khartoum State, Sudan

Nada Abdelmagid[1,2]*, Omama Abdalla[3], Abdallah Yagoub[3], Abeer Taha[3], Altayeb Hyder[3], Awatif Yahia[3], Bashar Hassan[3], Marwa Ali[3], Mohammed Abdeen[3], Mustafa Adam[3], Nadeen Kamal[3], Nader Ezeideen[3], Osama Altib[3], Rahma MohamedSalih[3], Salma Alnour[3], Sana Koko[3], Shama Abdelatif[3], Waleed Yahia[3], Yousif Abdelhade[3], Jennifer Palmer[2,4], Michelle Lokot[1,2], Bayard Roberts[1,2]

**1** Department of Health Services Research and Policy, Faculty of Public Health and Policy, The London School of Hygiene & Tropical Medicine, London, United Kingdom, **2** Health in Humanitarian Crises Centre, The London School of Hygiene & Tropical Medicine, London, United Kingdom, **3** Y-PEER Sudan, Khartoum, Sudan **4** Department of Global Health and Development, Faculty of Public Health and Policy, The London School of Hygiene & Tropical Medicine, London, United Kingdom

☯ These authors contributed equally to this work.
* nada.abdelmagid@lshtm.ac.uk

## Abstract

The frequency and severity of disease outbreaks disproportionately impacts settings affected by conflict or with weak health systems. Sudan, facing frequent and recurrent epidemics, struggles with limited resources. Understanding local perceptions of epidemic-prone diseases is vital for designing effective epidemic responses. This study describes the salience and perceptions of epidemic-prone diseases in two urban Sudanese communities. We conducted a cross-sectional qualitative study using freelisting in two communities in Khartoum State: Om Doum in Sharg al Neel locality and a neighbourhood in the Sixth Quarter of Ombadda locality. We purposively selected and interviewed consenting adults and recorded their responses to seven freelisting prompts. We analysed the data using Smith's salience index in Excel to evaluate the frequency and importance of terms, and used a relative salience index to compare terms between lists and sites. We interviewed 32 people in Om Doum and 30 in Ombadda. Epidemic-prone diseases, especially malaria, nose, sinus and throat infections (NSTIs), typhoid and COVID-19, were highly salient as common illnesses, recent outbreaks, and future infections and threats. Cancer and chronic diseases, while less salient, were important. Diseases highly salient as future threats, such as COVID-19 and cancer, were less salient, as likely future infections. Conversely, diseases highly salient as future infections, such as NSTIs and malaria, held lower salience for their future impact. This pattern was more pronounced in Ombadda. Infection prevention measures such as environmental hygiene were highly salient in both sites. Epidemic-prone diseases consistently emerged as significant

**Data availability statement:** All relevant data are within the manuscript and its Supporting Information files.

**Funding:** This work was supported by the Centers for Disease Control and Prevention (CDC) of the U.S. Department of Health and Human Services (HHS) [U01GH002319]. The contents are those of the author(s) and do not necessarily represent the official views of, nor an endorsement, by CDC/HHS or the U.S. Government. The CDC/HHS/U.S. Government had no role in the design of the study, the collection, analysis, and interpretation of data, or the writing of the manuscript.

**Competing interests:** The authors have declared that no competing interests exist.

concerns across sites, with local disease patterns shaping, but not fully explaining, perceived risks. While emphasising local disease burdens in risk communication is important, it may not be sufficient in all sites or for all diseases. Freelisting is useful for rapidly capturing the salience and perceptions of epidemic-prone diseases but requires complementary methods to explore nuanced patterns.

## Introduction

The twenty-first century has seen larger and more severe disease outbreaks, with the impact on mortality, morbidity and loss of livelihoods significantly greater in low and middle-income settings [1]. In these settings, concomitant vulnerabilities driven by extreme weather events and armed conflict increase the risk and impact of deadly epidemics [2,3]. A review of infectious disease outbreaks collected from the Disease Outbreak News and the Coronavirus Dashboard produced by the World Health Organization between 1996 and 2001 identified that the world's largest spatial cluster of outbreaks is in Africa [4]. Another analysis of infectious disease hotspots concluded that 22 of 25 epidemic-vulnerable countries in the world are in Africa [3].

Epidemics and outbreaks in many African countries are increasing in frequency and are often concurrent or simultaneous. They are driven by inadequate epidemic prevention and response, vaccination and other health services, among other factors [5,6]. While concurrent epidemics offer opportunities for pooling resources and strengthening coordinated responses, responders face immense difficulties scaling up responses and containing epidemics early.

Sudan is no exception to this situation. In 2020, polio, dengue fever, chikungunya, malaria and COVID-19 outbreaks were reported [7]. Similarly, in 2021 and 2022, COVID-19, hepatitis E, dengue fever, malaria, measles and monkeypox outbreaks were responded to by the Ministry of Health and its partners [8]. While Sudan has attempted to harmonise epidemic preparedness and response in a national 'multi-hazard' approach that coordinates the work of the government, relevant UN agencies, and national and international NGOs, preparedness and response are chronically hampered by financial and operational constraints posed by competing priorities [9,10]. The outbreaks and outbreak responses occur in the context of chronic conflict and insecurity, high exposure to natural hazards, social and economic fragility and high humanitarian needs. Since the eruption of large-scale armed conflict in April 2023, humanitarian needs have escalated dramatically while population and health system coping capacities have been severely eroded [11].

In Sudan, communities are often the first and primary responders to crises, including to overlapping and concurrent outbreaks [12–14]. For humanitarian and health sectors to support local responders, understanding how populations recognise epidemic-prone diseases and epidemics and perceive their associated risks within local social contexts is crucial for designing effective and participatory epidemic responses, improving risk communication, fostering trust in professional responses, and ensuring a coordinated and culturally appropriate response to health crises

[15,16]. However, locally constructed understandings of epidemic-prone diseases and epidemic experiences in Sudan are poorly documented [17]. This study aimed to describe the salience of epidemic-prone diseases among members of two urban Sudanese communities and describe how they assess associated risks.

## Materials and methods

### Ethics statement

The study received approval by Sudan's National Health Research Ethics Committee (number 1-5-22) and the London School of Hygiene and Tropical Medicine's Research Ethics Committee (number 26466). The study was conducted according to the principles expressed in the Declaration of Helsinki. Written informed consent was obtained from all participants.

### Study design

We conducted a cross-sectional qualitative study. We conducted individual interviews with eligible consenting participants using freelisting. Freelisting is an elicitation technique traditionally used in anthropological research to define the elements of a given cultural domain from the insider perspective of a cultural group's member [18]. The technique entails asking research participants to list all items they believe belong in a category. The frequency and order of items in a list are considered indicators of the salience of these items within a domain. In our study, the cultural domain was "epidemic-prone diseases". As authors, we define epidemic-prone diseases as infectious diseases that have the potential to cause outbreaks or epidemics due to their ability to spread rapidly within populations. However, the freelisting technique avoids researchers' presupposed or imposed definitions and categorisations of the cultural domain. Freelisting is particularly useful in providing results for incorporation into mixed methods studies in the same community [19]. It has been previously used to explore local understandings or perceptions of illnesses in similar settings [20–23]. Nonetheless, freelisting has some limitations. The technique relies on memory, and there is evidence that participants may be more inclined to list recent or familiar items and may miss others due to poorly framed or unclear freelisting prompts [24,25]. Furthermore, freelisting responses are usually individual words or short phrases, lacking the deeper insights that come from interviews and focus group discussions [19].

The study used a participatory research approach, engaging youth volunteers from Khartoum, some of whom lived in the study communities. The volunteers are members of Sudan's Youth Peer Education Network (Y-PEER Sudan), a nationwide network of trained youth volunteers active in promoting health and youth participation. The first author, NA, and youth volunteers from Y-PEER Sudan are members of the Sudan Research Group, which had previously conducted participatory research projects in Sudan [26–28]. The youth volunteers participated in identifying and selecting study sites, designing the data collection instruments and conducting data analysis. Participatory research seeks to afford equal legitimacy to scientific knowledge held by 'experts', and experiential, embodied and affective knowledge held by 'lay' people [29].

### Study setting

We conducted the study in two communities in Khartoum State, Sudan. The state includes the national capital city, hosts both state- and national-level health authorities, has a thriving private health sector and is home to many tertiary and specialised medical services. Khartoum State has a recent history of several outbreaks, including malaria, COVID-19, cholera, and Crimean-Congo Haemorrhagic Fever, as well as environmental evidence of vaccine-derived poliovirus circulation [30–33]. At the time of data collection in 2022, there were ongoing malaria, COVID-19 and monkeypox outbreaks in Khartoum [34–36]. On 15th April 2023, a large-scale armed conflict erupted in Khartoum and spread to many parts of Sudan, with massive large-scale population displacement, including in both study sites.

The choice of the study locations was primarily based on operational criteria, considering factors such as the presence of youth volunteers and the communities' inclination and interest to be part of the study, as well as being in different localities in Khartoum and have different socioeconomic contexts, as described below.

Our first site is in Sharg al Neel locality, east of the Blue Nile River in Khartoum State. It covers 9,385 square kilometres and had up to 1.1 million residents before the 2023 conflict, including displaced persons and rural and urban populations. It hosts large agricultural schemes and experiences frequent seasonal flash flooding. Om Doum, a town on the river's banks, had a population of about 40,000 before the conflict and traces its roots to 15th-century settlers. We focused on the "old" central part of Om Doum, the original settlement that expanded into the current town. Om Doum is relatively geographically isolated within Sharg Alneel, and has a distinctive social composition with urban features but a rural social feel. At the time of our study, Om Doum was a close-knit community with strong familial and social ties, most of whom are linked to the original town settlers. Households were multi-generational, typically with 8–10 members, and most relied on petty or lucrative trades. Wealth varied, reflected in housing ranging from small bungalows to large mansions. Homes had electricity and water from Khartoum's grid. The town had one sizeable open-air market,13 schools, and one health centre and was connected to a main road with reasonable transportation links. Predominantly Muslim, residents follow Sufi traditions and speak a colloquial Sudanese Arabic dialect. There is a local artisanal brickmaking industry in Om Doum, with the labourers primarily being South Sudanese refugees and the owners being wealthy industrialists from other parts of Khartoum. The refugees reside in informal settlements on the peripheries of Om Doum and have little interaction with our study community.

Our second site is in the Ombadda locality, a densely populated area in western Khartoum State, hosting up to 2.5 million low- and middle-income urban, displaced and rural residents before April 2023. Within Ombadda's urban Sixth Quarter (*alhara alsadsa*) we selected a southern neighbourhood, covering about 0.3 square kilometres with around 1,800 residents, most of whom had lived there for at least two generations, and included newcomers marrying into resident families. Households were typically 5–10 nuclear and extended family members residing in single-storey homes with electricity and water from Khartoum's grid. Most residents earned from low-wage jobs, service brokerage, or small-scale trades, with most educated to at least secondary school level. The area has good transport links, nearby private health services, and schools. The predominantly Muslim population stems from various Sudanese tribes, speaking a colloquial Sudanese Arabic dialect.

## Study participants

We interviewed eligible consenting adults in each study community. Eligible participants were persons aged 18 years or more, who were residing in the study sites for at least three years and who agreed to provide written informed consent to participate in the study. We identified participants purposively, guided by the study site coordinators (co-authors M Abdeen and YA), who were community members. In most instances, we identified participants already convened at a local event (e.g., a health education campaign), a social gathering (e.g., women's coffee gathering) or in a public space (e.g., a barber shop). Each participant was also asked to suggest another eligible participant for the interview. We aimed to interview at least 20 participants per study site as more interviews are not likely to yield unique data in homogenous groups [19,37]. Study site coordinators and data collectors ensure diversity in the age and gender of recruited participants through coordinated recruitment and communication throughout the 2 days of data collection in each site.

## Data collection

Co-author NA developed seven preliminary prompts. During a training workshop for youth volunteers, we developed a lexicon of the key study concepts (epidemics, epidemic-prone diseases, future risk) in the colloquial Sudanese dialect dominant in the two study communities. A detailed discussion, review and rephrasing of the draft data collection tool followed this. The freelisting prompts were pre-tested within the workshop setting through role-playing. The final data collection

tool consisted of seven prompts covering four themes: (i) the extent to which epidemic-prone diseases feature among the main health concerns of participants, (ii) the extent to which recent epidemics are recognised or acknowledged by participants, (iii) risk perceptions of specific epidemic-prone diseases: perceived likelihood and severity, and (iv) communication about epidemic-prone disease and epidemics within the community (S1 Text).

Trained youth volunteers held in-person interviews between November 26th and 28th, 2022, after participants were invited and written informed consent was obtained and documented on a form. After obtaining participants' permission, interviews were audio recorded. Two participants in Om Doum declined audio recording, and the interviewer wrote their responses by hand. As most interviews took place in a public space, care was taken to ensure that participants could not overhear other ongoing interviews and be influenced in their responses. The interviews ranged from 3 to 11 minutes in Om Doum, and 3 to 17 minutes in Ombadda.

### Data analysis

We analysed data in two stages. We did a preliminary analysis immediately after data collection to inform further research in both study sites. The analysis was done in a workshop that started by listing all the terms mentioned by participants in responses to prompts 4 and 6, followed by a discussion amongst the youth volunteers, which included members of the study communities, to group synonyms and similar terms. We first merged all singular and plural forms of the term, then merged synonyms, and finally merged similar "concepts" (S2 Text). We used Microsoft Excel to calculate the salience index of each term in each list (Smith's $S$). Smith's salience index takes into account both the frequency of mention of an item across the lists and the rank of the item in the lists [38]. Following the verbatim transcription of audio-recorded interviews, we conducted a complete analysis of the remaining prompts.

We developed a relative salience index to compare terms between lists and sites. This was calculated by dividing each term's raw salience index by the raw index of the most salient term in its list. We named this a "normalised salience index". Comparisons were restricted to the most salient terms in each list. The cut-offs for determining the most salient terms were based on two criteria: either the top terms before an inflexion point in a scree plot of salience scores or, in cases where no clear inflexion point was present, or multiple inflexions were observed, the top 5 salient terms were selected.

All the analysis was conducted in Arabic. Translation of terms into English was done by NA after all concepts were merged and final lists for each prompt were compiled. Translations were cross-checked by OA.

### Results

We interviewed 32 people in Om Doum and 30 in Ombadda. The characteristics of study participants are presented in Table 1. A few respondents were not correctly asked or did not respond to some prompts; respondents per prompt ranged from 29 to 32 in Om Doum, and from 28 to 30 in Ombadda. The average length of freelists in both sites ranged from 2 to 4 terms per list.

Below we present the salient terms by the following themes: (i) the extent to which epidemic-prone diseases feature among the main health concerns of participants (prompts 1, 2 and 3), (ii) the extent to which recent epidemics are recognised or acknowledged by participants (prompt 4), (iii) risk perceptions of specific epidemic-prone diseases: perceived likelihood and severity (prompts 5 and 7), and (iv) communication about the epidemic-prone disease and epidemics (prompt 6). Scree plots of the most salient terms in each list are available in S3 Text. We also present similarities and differences between the most salient terms in freelists; we compared salient terms for prompts 1 and 3, 1 and 4, 5 and 7, 1 and 5, and 1 and 7. We provide illustrative quotes to contextualise negative responses.

### The extent to which epidemic-prone diseases feature among the main health concerns of participants

Infectious diseases, including epidemic-prone ones, emerged as the most salient reported common illnesses in respondents' neighbourhoods and as illnesses they frequently heard about (Table 2). Malaria, nose, sinus and throat infections

**Table 1. Characteristics of study participants.**

| Characteristics of study participants | Om Doum *(n = 32)* | Ombadda *(n = 30)* |
|---|---|---|
| **Age** | | |
| Range (years) | 19 – 97 | 21 – 64 |
| Average (years) | 32 | 37 |
| **Gender** | | |
| Woman *(n, %)* | 18 (60%) | 14 (47%) |
| Man *(n, %)* | 14 (47%) | 16 (53%) |
| **Occupation** | | |
| Tradesperson/Skilled Laborers *(n, %)* | 5 (16%) | 2 (7%) |
| Small business owner *(n, %)* | 0 (0%) | 2 (7%) |
| Homemaker *(n, %)* | 3 (9%) | 12 (40%) |
| Student *(n, %)* | 13 (41%) | 5 (17%) |
| Daily worker/service broker *(n, %)* | 2 (6%) | 5 (17%) |
| Health worker *(n, %)* | 1 (3%) | 1 (3%) |
| Professionals and Managers *(n, %)* | 3 (9%) | 3 (10%) |
| Unemployed *(n, %)* | 2 (6%) | 0 (0%) |
| Unknown *(n, %)* | 3 (9%) | 0 (0%) |
| **Number of years residing in the study site** | | |
| Range (years) | 3 – 97 | 3 – 64 |
| Average (years) | 23 | 23 |

(NSTI), typhoid and COVID-19 appeared in the top five salient illnesses in both locations. However, cancer and chronic diseases like diabetes and hypertension were also mentioned. When respondents listed health issues that mattered to them, prevention measures against salient infectious diseases featured prominently in both sites. Environmental hygiene held the highest salience term in both locations. Access to health services and information emerged as less salient but crucial factors.

### The extent to which recent epidemics are recognised or acknowledged by participants

Om Doum and Ombadda participants reported recent outbreaks in their neighbourhoods over the past three years (Table 3). COVID-19, malaria, and NSTI emerged as the most salient, with COVID-19 holding equal salience in both sites. Respondents in both sites also mentioned non-infectious causes including cancer, chronic diseases, and allergies. Notably, the term "No outbreaks" received a low salience score in both locations

### Perceived likelihood and severity of epidemic-prone diseases

Prompt 5 refers to the perceived odds or chances of personal infection in the next year whereas Prompt 7 refers to the perceived effects of an infectious disease in the next year (which can be interpreted at a personal, family, community or wider level; in a positive or negative light; and in health, economic, social or aspects of life). When asked to anticipate infectious diseases they might contract in the coming year, and, separately, that might affect them in the next year, respondents reported epidemic-prone infections that featured prominently in the earlier freelists (Table 4). Malaria, COVID-19, and NSTI were highly salient in both lists and in both study sites.

Reluctance to predict was minimal or absent in Om Doum, while in Ombadda, it was highly salient. Respondents justified their reluctance to predict by mentioning cultural and faith-related perspectives that discourage predictions.. Respondents also mentioned cancer, chronic diseases, and allergies.

**Table 2. Compound salience indices, by site, of all responses to prompts 1, 2 and 3: "List all the common illnesses in your neighbourhood", "List all the illnesses that you hear about", "List all the health issues that matter to you.".**

| Om Doum | | | | | | Ombadda | | | | | |
|---|---|---|---|---|---|---|---|---|---|---|---|
| List all the common illnesses in your neigh-bourhood | Com-pound salience index (n=32) | List all the illnesses that you hear about | Com-pound salience index (n=31) | List all the health issues that matter to you | Com-pound salience index (n=31) | List all the common illnesses in your neigh-bourhood | Com-pound salience index (n=30) | List all the illnesses that you hear about | Com-pound salience index (n=30) | List all the health issues that mat-ter to you | Com-pound salience index (n=30) |
| Malaria | 0.95 | Malaria | 0.65 | Clean environment | 0.29 | Malaria | 0.71 | Malaria | 0.52 | Clean environment | 0.50 |
| Nose, sinus and throat infections (NSTI)* | 0.42 | Typhoid | 0.31 | Vector control (mosquito) | 0.24 | Nose, sinus and throat infections (NSTI)* | 0.41 | Cancer | 0.34 | Rubbish disposal | 0.19 |
| Typhoid | 0.33 | COVID-19 | 0.28 | Eliminate stagnant water | 0.20 | Typhoid | 0.29 | Typhoid | 0.28 | Eliminate stagnant water | 0.17 |
| COVID-19 | 0.14 | Nose, sinus and throat infections (NSTIs)* | 0.20 | Personal health and hygiene | 0.15 | Chronic diseases | 0.11 | Nose, sinus and throat infections (NSTIs)* | 0.23 | Healthy eating | 0.12 |
| Diarrhoeas | 0.09 | Cancer | 0.18 | Health promotion | 0.12 | Cancer | 0.10 | COVID-19 | 0.21 | Nearby health services | 0.08 |
| Cholera | 0.08 | Chronic diseases | 0.17 | Availability of nearby hospitals | 0.11 | COVID-19 | 0.07 | Chronic illnesses | 0.16 | Vector control (mosquito) | 0.08 |
| Chronic diseases | 0.08 | Fevers | 0.11 | Health diag-nostic services | 0.06 | Stomach bug† | 0.06 | Dengue fever | 0.12 | Compre-hensive and quality health services | 0.05 |
| Influenza | 0.03 | Stomach bug† | 0.09 | Climate change | 0.05 | Diarrhoeas | 0.05 | Stomach bug† | 0.11 | Clean water | 0.05 |
| Jaundice | 0.03 | Diarrhoeas | 0.09 | Availability of medical pro-fessionals and equipment | 0.04 | Fevers | 0.05 | Hepatitis | 0.11 | Health promotion | 0.04 |
| Urinary infections | 0.03 | Cholera | 0.08 | Clean food | 0.03 | Influenza | 0.05 | Fevers | 0.08 | Affordable health services | 0.04 |
| Fevers | 0.03 | Hepatitis | 0.04 | Rapid response by Ministry of Health | 0.03 | Urinary infections | 0.03 | Diarrhoeas | 0.06 | Collective community health action | 0.04 |
| Cancer | 0.02 | Allergies | 0.03 | Free treatment | 0.03 | Headaches | 0.03 | Chest/lung infection | 0.06 | Prevention of chronic illnesses | 0.03 |
| Dysentery | 0.02 | Influenza | 0.02 | Availability of vaccines | 0.03 | Epidemics | 0.03 | Bone diseases | 0.05 | Staying at home | 0.03 |
| Stomach bug† | 0.02 | Dysentery | 0.02 | None | 0.03 | Anaemia | 0.02 | Kidney and urinary diseases | 0.04 | None mentioned | 0.03 |

*(Continued)*

**Table 2.** (Continued)

| Om Doum | | | | | | Ombadda | | | | | |
|---|---|---|---|---|---|---|---|---|---|---|---|
| List all the common illnesses in your neighbourhood | Compound salience index (n=32) | List all the illnesses that you hear about | Compound salience index (n=31) | List all the health issues that matter to you | Compound salience index (n=31) | List all the common illnesses in your neighbourhood | Compound salience index (n=30) | List all the illnesses that you hear about | Compound salience index (n=30) | List all the health issues that matter to you | Compound salience index (n=30) |
| Anaemia | 0.02 | Infections other than nose, sinus and throat infections | 0.02 | Saline water | 0.02 | Cholera | 0.01 | Cholera | 0.04 | Livelihoods | 0.03 |
| | | Slipped disc | 0.02 | Healthy food | 0.02 | | | AIDS | 0.04 | COVID-19 prevention | 0.03 |
| | | Irritable colon‡ | 0.02 | Mental support to patients | 0.02 | | | Headache | 0.03 | | |
| | | | | | | Middle ear infection | 0.03 | | | | |
| | | | | | | Anaemia | 0.02 | | | | |
| | | | | | | Monkey pox | 0.02 | | | | |
| | | | | | | Allergies | 0.02 | | | | |
| | | | | | | White water in eyes | 0.02 | | | | |
| | | | | | | Skin diseases | 0.01 | | | | |
| | | | | | | Blood infection | 0.01 | | | | |
| | | | | | | Yellow fever | 0.01 | | | | |

*Nose, sinus and throat infections (NSTI)*: this category includes terms that refer to common colds, sinus infections and sore throats.

† Stomach bug is a colloquial term in the study areas that refers to *H. pylori* infection.

‡ Irritable colon is a colloquial term in the study areas that refers to irritable bowel syndrome.

*"The future is unknown. I can't predict what (infection) will come to me"* – 50-year-old man, Ombadda

- Interviewer: *"List all the infectious diseases that you can get in the next year"*

- Respondent: *"May Allah not bring them to us; why are you bringing them to us [by asking this question]?"* 40-year-old man, Ombadda

### Receiving information and communicating about epidemic-prone diseases

When asked to list the places (social and physical) where they discuss outbreaks, responses from Om Doum participants revealed that most outbreak-related communication occurs within the confines of the neighbourhood and its population: the local health centre and nursery/kindergarten, local friends and coffee gatherings, and at home (Table 5). In Om Doum, this includes strong familial connections amongst Om Doum residents (see "Methods"). Communication included lay people and local health professionals. The salient terms suggest that communication occurs among individuals within the community and with less intimate connections during social events.

In contrast, responses from Ombadda indicated that communication about outbreaks occurs within the neighbourhood and the broader district, extending to places like markets and hospitals (Table 5). The most prominent concepts suggest that communication in Ombadda occurs mainly among lay people, and less so with health professionals.

**Table 3. Compound salience indices, by site, of all responses to prompt 4: "List all the outbreaks that occurred in your neighbourhood in the past 3 years".**

| Om Doum | | Ombadda | |
|---|---|---|---|
| **List all the outbreaks that occurred in your neighbourhood in the past 3 years** | **Compound salience index (n = 32)** | **List all the outbreaks that occurred in your neighbourhood in the past 3 years** | **Compound salience index (n = 30)** |
| COVID-19 | 0.53 | COVID-19 | 0.39 |
| Malaria | 0.52 | Nose, sinus and throat infections (NSTI)* | 0.33 |
| Nose, sinus and throat infections (NSTI)* | 0.20 | Malaria | 0.25 |
| Typhoid | 0.11 | Stomach bug† | 0.12 |
| Diarrhoeas | 0.08 | Typhoid | 0.11 |
| Cholera | 0.06 | Cancer | 0.10 |
| Cancer | 0.06 | Tuberculosis | 0.10 |
| Fevers | 0.05 | Chronic diseases | 0.05 |
| Allergies | 0.05 | Hepatitis | 0.04 |
| Chronic diseases | 0.04 | Cholera | 0.03 |
| Measles | 0.04 | Chicken pox | 0.03 |
| Influenza | 0.03 | None | 0.03 |
| Chicken pox | 0.03 | Dengue fever | 0.03 |
| None | 0.03 | Intestinal infections | 0.03 |
| Dysentery | 0.02 | Fevers | 0.02 |
| Mumps | 0.01 | Allergies | 0.02 |
| Stomach bug† | 0.00 | Urinary infections | 0.01 |

*Nose, sinus and throat infections (NSTI): this category includes terms that refer to common colds, sinus infections and sore throats.

† Stomach bug is a colloquial term in the study areas that refers to *H. pylori* infection.

### Similarities and differences in salient terms between freelists

**Comparing common illnesses (prompt 1) and illnesses that respondents hear about (prompt 3).** In both Om Doum and Ombadda, there is a noticeable alignment between the salient common illnesses reported in the neighbourhood and the illnesses participants reported hearing about (Fig 1). Malaria, NSTI, and typhoid were highly salient on both lists.

The normalised salience indices for malaria and NSTI are similar between lists in both sites, indicating a shared emphasis. However, a nuanced pattern emerges with other illnesses. For typhoid in both sites and COVID-19 in Om Doum, the indices are lower when considered as common illnesses than when participants report hearing about them. This suggests that these diseases might be perceived as more noteworthy than their actual prevalence within the neighbourhoods.

Notably, cancer is highly salient in both sites as an illness participants heard about, but it does not rank among the most common illnesses. The same applies to COVID-19 in Ombadda. This finding raises the possibility that discussions in the population may prioritise "severe" illnesses, even if they are not perceived as highly common in the neighbourhoods.

**Comparing common illnesses (prompt 1) and outbreaks that occurred in the past three years (prompt 4).** In both Om Doum and Ombadda, malaria and NSTI were highly salient as recent outbreaks and as common illnesses (Fig 2). In Om Doum, normalised salience indices for malaria and NSTI are similar between the two lists, while in Ombadda the indices varied markedly between the lists. These findings suggest diverse interpretations of the concepts of "outbreaks" and "common illnesses", with Ombadda respondents making a stronger distinction between the two concepts.

In both sites, COVID-19 had a higher normalised salience index as a reported outbreak than as a common illness. This may be influenced by the timing of data collection and phrasing of the prompts, as local COVID-19 cases had decreased at the time of the study. Still, it had received significant public attention in the three years preceding data collection.

**Table 4. Compound salience indices, by site, of all responses to prompts 5 and 7: "List all the infectious diseases that you can get in the next year" and "List all the infectious diseases that can affect you in the next year".**

| Om Doum | | | | Ombadda | | | |
|---|---|---|---|---|---|---|---|
| List all the infectious diseases that you can get in the next year | Compound salience index (n=29) | List all the infectious diseases that can affect you in the next year | Compound salience index (n=30) | List all the infectious diseases that you can get in the next year | Compound salience index (n=29) | List all the infectious diseases that can affect you in the next year | Compound salience index (n=30) |
| Malaria | 0.47 | Malaria | 0.46 | Nose, sinus and throat infections (NSTI)* | 0.22 | COVID-19 | 0.33 |
| Nose, sinus and throat infections (NSTI)* | 0.26 | COVID-19 | 0.23 | (Reluctant to predict)§ | 0.21 | Malaria | 0.22 |
| COVID-19 | 0.19 | Nose, sinus and throat infections (NSTI)* | 0.22 | Malaria | 0.19 | (Reluctant to predict)§ | 0.20 |
| Chronic diseases | 0.12 | Typhoid | 0.17 | COVID-19 | 0.18 | Dengue fever | 0.12 |
| Diarrhoeas | 0.10 | Cancer | 0.16 | Typhoid | 0.10 | Nose, sinus and throat infections (NSTIs)* | 0.10 |
| Typhoid | 0.08 | Chronic diseases | 0.12 | Tuberculosis | 0.09 | Tuberculosis | 0.09 |
| Fevers | 0.06 | Allergies | 0.08 | Dengue fever | 0.07 | Cancer | 0.07 |
| Cancer | 0.05 | Chicken pox | 0.05 | Cancer | 0.05 | Chronic illnesses | 0.07 |
| Influenza | 0.03 | Measles | 0.04 | Chronic diseases | 0.03 | Cholera | 0.07 |
| Hepatitis | 0.03 | Cholera | 0.03 | Cholera | 0.03 | Hepatitis/jaundice | 0.06 |
| (Reluctant to predict)§ | 0.03 | Dengue fever | 0.03 | Influenza | 0.03 | Intestinal infections | 0.04 |
| Stomach bug† | 0.02 | Fractures | 0.03 | Bone diseases | 0.03 | Typhoid | 0.04 |
| Allergies | 0.02 | Irritable colon‡ | 0.03 | Allergies | 0.03 | Skin diseases | 0.03 |
| | | Immunity diseases | 0.03 | Headache | 0.03 | Chest/lung infection | 0.03 |
| | | Meningitis | 0.02 | AIDS | 0.03 | Allergies | 0.03 |
| | | Sexually-transmitted infections | 0.02 | Dysentery | 0.03 | New/novel diseases | 0.03 |
| | | Hepatitis | 0.02 | Fevers | 0.02 | Bilharziasis | 0.03 |
| | | Stomach bug† | 0.02 | Hepatitis/jaundice | 0.02 | Stomach bug† | 0.03 |
| | | Slipped disc | 0.01 | Urinary infections | 0.02 | Dysentery | 0.02 |
| | | | | Stomach bug† | 0.01 | Monkey pox | 0.01 |
| | | | | | | Malnutrition | 0.01 |

*Nose, sinus and throat infections (NSTI): this category includes terms that refer to common colds, sinus infections and sore throats.

† Stomach bug is a colloquial term in the study areas that refers to *H. pylori* infection.

‡Irritable colon is a colloquial term in the study areas that refers to irritable bowel syndrome.

§"Reluctant to predict" refers to responses where participants declined to answer these prompts, citing religious or cultural beliefs that discourage predictions about the future.

The findings above suggest that the study populations recognised epidemics through the observed behaviour of the disease within the community, e.g., seeing malaria or NSTI cases increase rapidly or occur in clusters, and through official declarations of an outbreak, such as with COVID-19.

**Comparing predicted infections acquired (prompt 5) and predicted infections that could affect respondents (prompt 7).** Malaria, NSTI, and COVID-19 were consistently present in both Om Doum and Ombadda lists, suggesting

**Table 5. Compound salience indices, by site, of all responses to prompt 6: "List all the places (social and physical) where you talk about infectious diseases".**

| Om Doum | | Ombadda | |
|---|---|---|---|
| List all the places (social and physical) where you talk about infectious diseases | Compound salience index (n = 31) | List all the places (social and physical) where you talk about infectious diseases | Compound salience index (n = 28) |
| Health centre | 0.33 | Neighbourhood meetings/gatherings | 0.28 |
| Friends | 0.17 | At home | 0.20 |
| At home | 0.17 | Women's gatherings | 0.17 |
| Coffee gatherings | 0.15 | At the hospital | 0.11 |
| Nursery/kindergarten | 0.13 | Markets and public places | 0.10 |
| Social events (funerals or celebrations) | 0.09 | Radio and TV | 0.10 |
| The street and public transport | 0.09 | Coffee gatherings | 0.10 |
| With pharmacists or doctors | 0.09 | Between family members | 0.09 |
| School | 0.09 | Social events (weddings and funerals) | 0.09 |
| Neighbourhood social media platforms | 0.08 | Mosques | 0.09 |
| Awareness campaigns | 0.06 | Social media platforms | 0.08 |
| Social home visits | 0.05 | Between neighbours | 0.08 |
| Mosques | 0.05 | At the university | 0.07 |
| Neighbourhood committees or associations | 0.05 | Social home visits | 0.05 |
| Media | 0.04 | At the pharmacy | 0.04 |
| University | 0.04 | Between friends | 0.03 |
| Clubs | 0.03 | At clubs | 0.03 |
| Neighbourhood square | 0.03 | With health professionals | 0.02 |
| Market | 0.03 | (Young) men's gatherings | 0.02 |
| Cafe/restaurant | 0.03 | At the nursery | 0.02 |
| With neighbours | 0.03 | On the street and public transport | 0.01 |
| Hospitals | 0.01 | | |

these are shared concerns. The normalised salience indices of malaria, NSTI, and COVID-19 in the two lists show a similar pattern in both sites, but the exaggeration is more pronounced in Ombadda (Fig 3). NSTI and malaria had a higher salience for probability but lower salience for their potential impact on the participant. In contrast, COVID-19 had a lower salience for probability than for its possible effect on the respondent. Beyond NSTI, COVID-19 and malaria, there were no other shared terms between the two lists.

Respondents in both locations seemed to differentiate between the perceived likelihood of contracting an illness and the perceived severity of that illness. This distinction is more pronounced in Ombadda. For instance, apart from malaria in Om Doum, respondents in both sites believed their chances of contracting more severe diseases like COVID-19, cancer, and dengue fever were lower, even though they perceived a higher severity of these diseases. Conversely, they perceived less severe illnesses, such as NSTI in both sites and malaria in Ombadda, as having a higher likelihood of occurrence.

**Comparing common illnesses (prompt 1), predicted infections acquired (prompt 5) and predicted infections that could affect respondents (prompt 7).** When comparing the normalised salience indices in Om Doum of common illnesses with predicted infections respondents might acquire in the next year, malaria, NSTI and chronic diseases consistently held similar salience in both lists (Fig 4). Similarly, when comparing the normalised salience indices of common illnesses with predicted infections that could affect respondents, malaria, NSTI, and typhoid demonstrated similar salience in both lists (Fig 5). These findings suggest a potentially strong influence of local disease epidemiology on the perceived probability of infection and disease severity among Om Doum respondents.

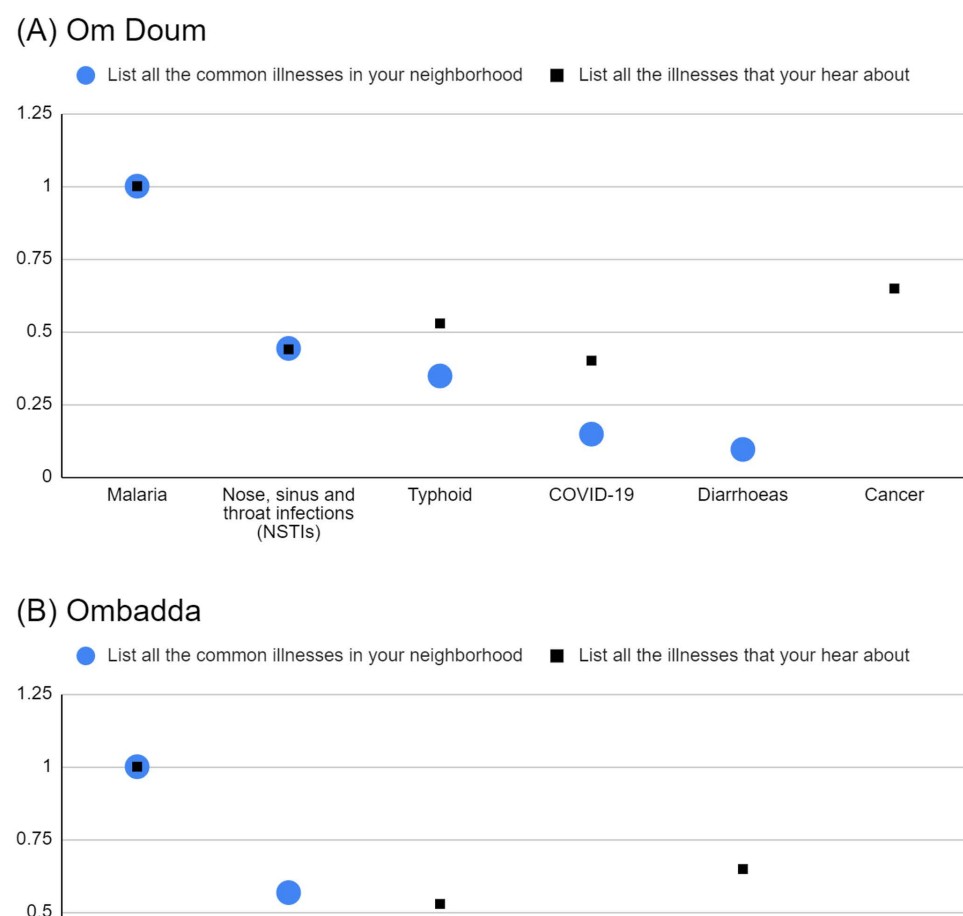

(A) Om Doum

● List all the common illnesses in your neighborhood    ■ List all the illnesses that your hear about

(B) Ombadda

● List all the common illnesses in your neighborhood    ■ List all the illnesses that your hear about

**Fig 1. Normalised salience indices of salient common illnesses (prompt 1) and salient illnesses that respondents heard about (prompt 3) in Om Doum (A) and Ombadda (B).**

The findings in Ombadda reveal a more nuanced perspective. While NSTI, malaria, and typhoid appeared as common illnesses and predicted infections respondents might acquire in the next year, discrepancies in their salience between lists were observed (Fig 4) Malaria and NSTI had opposing saliences between lists, while typhoid held similar salience in both lists. Similarly, when comparing the normalised salience indices of common illnesses with predicted infections that could affect respondents, only malaria and NSTI appeared in both lists, and both showed lower salience indices for potential effect next year than as commonly reported illnesses (Fig 5). These findings suggest that local disease epidemiology was not consistently influential on Ombadda participants when gauging the probability of infection and severity of disease and that its level of influence varies by disease.

Notably, COVID-19 and cancer in Om Doum and COVID-19 and dengue fever in Ombadda emerged as highly salient future illnesses that could affect participants despite not being prominently reported as common illnesses (Fig 5).

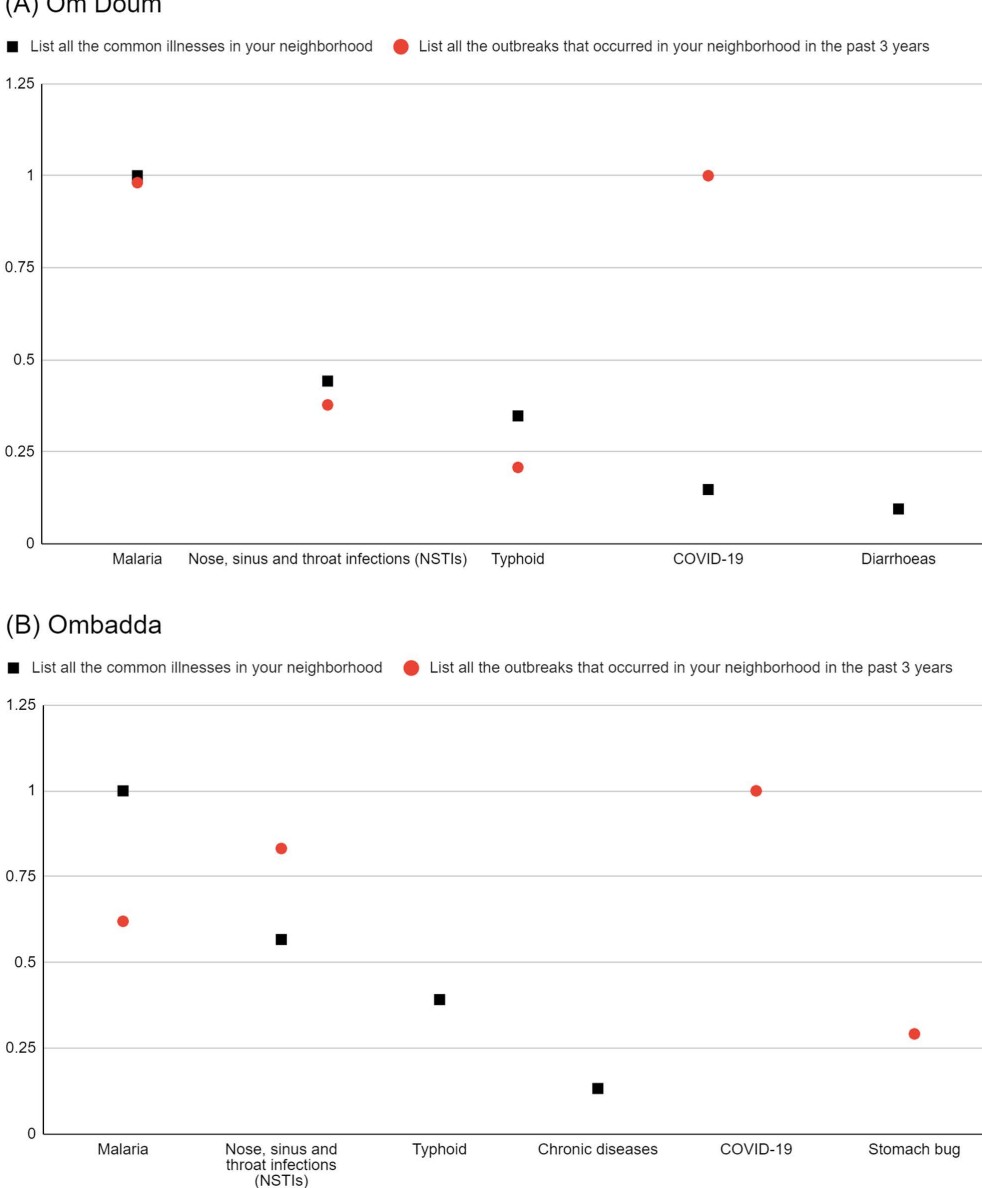

**(A) Om Doum**

**(B) Ombadda**

**Fig 2. Normalised salience indices of salient reported common illnesses (prompt 1) and salient reported recent outbreaks (prompt 4) in Om Doum (A) and Ombadda (B).**

Furthermore, in both Om Doum and Ombadda, COVID-19 had a higher salience as a potential future-acquired infection (Fig 4). These findings indicate that, even with reduced or absent local cases, participants did not discount the possibility of contracting COVID-19 or being affected by COVID-19, cancer or dengue fever in the coming year.

## Discussion

This is the first study in Sudan to explore the salience and perceptions of multiple epidemic-prone diseases among community members. Our findings reveal that these diseases are highly salient, with respondents recognising recent

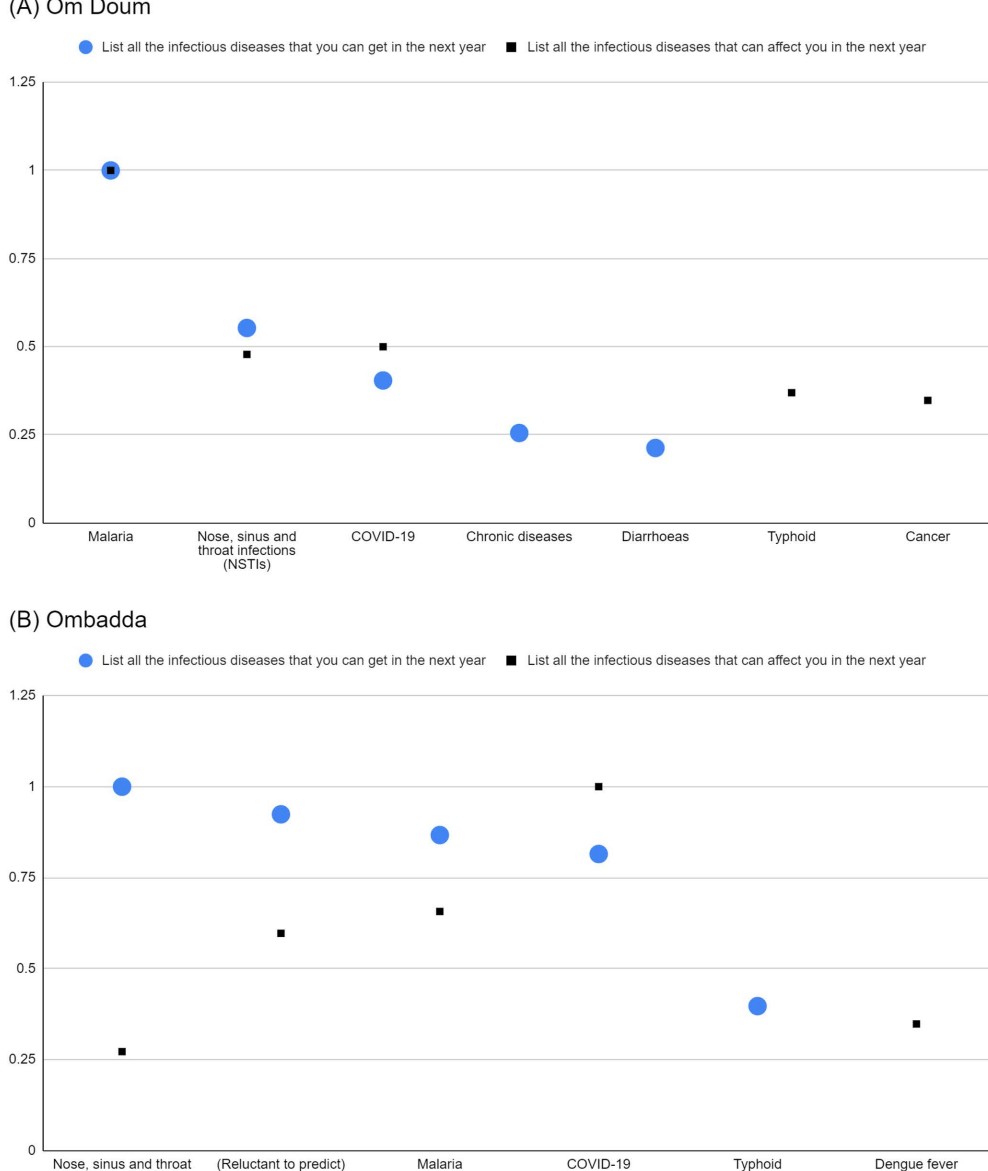

**Fig 3. Normalised salience indices of predicted infections acquired (prompt 5) and predicted infections that could affect respondents (prompt 7) in Om Doum (A) and Ombadda (B).**

outbreaks. Respondents differentiated between the likelihood of contracting an infection and its perceived severity. In addition, local disease patterns significantly influence risk perception, though they do not fully account for it. Freelisting demonstrated utility as a rapid method for identifying community perceptions of epidemic-prone diseases, but to gain deeper, actionable insights for designing risk communication interventions, it should be combined with complementary quantitative or qualitative methods in a mixed-methods design.

In both sites, epidemic-prone diseases, especially malaria, NSTI, typhoid and COVID-19, were consistently highly salient as common illnesses, recent outbreaks, potential future infections, and future threats to people's lives. Cancer and

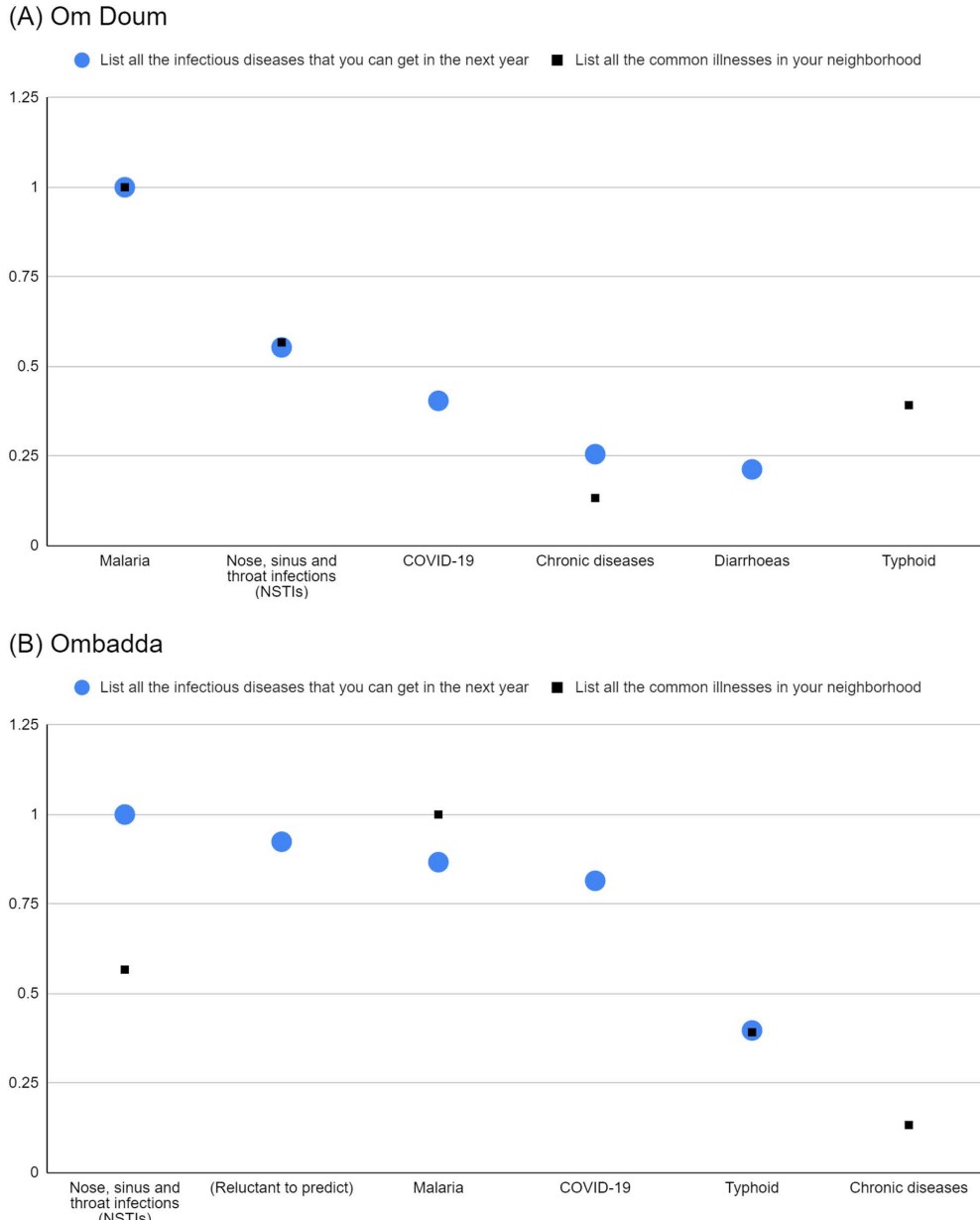

(A) Om Doum

● List all the infectious diseases that you can get in the next year  ■ List all the common illnesses in your neighborhood

(B) Ombadda

● List all the infectious diseases that you can get in the next year  ■ List all the common illnesses in your neighborhood

**Fig 4. Normalised salience indices of reported common illnesses (prompt 1) and predicted infections acquired (prompt 5) in Om Doum (A) and Ombadda (B).**

chronic diseases, while less salient, were also important, consistent with the reported rising burden of non-communicable and communicable diseases in Sudan [39]. According to the World Health Organization, in 2021, the leading causes of death in Sudan were communicable, maternal, perinatal, and nutritional conditions (44.1%), followed closely by noncommunicable diseases (43.3%), with injuries (8.7%) and COVID-19-related outcomes (4%) contributing smaller shares [40]. Major contributors included malaria, measles, and lower respiratory infections for communicable diseases; ischaemic heart disease, stroke, and congenital anomalies for noncommunicable diseases; and COVID-19 was a significant cause

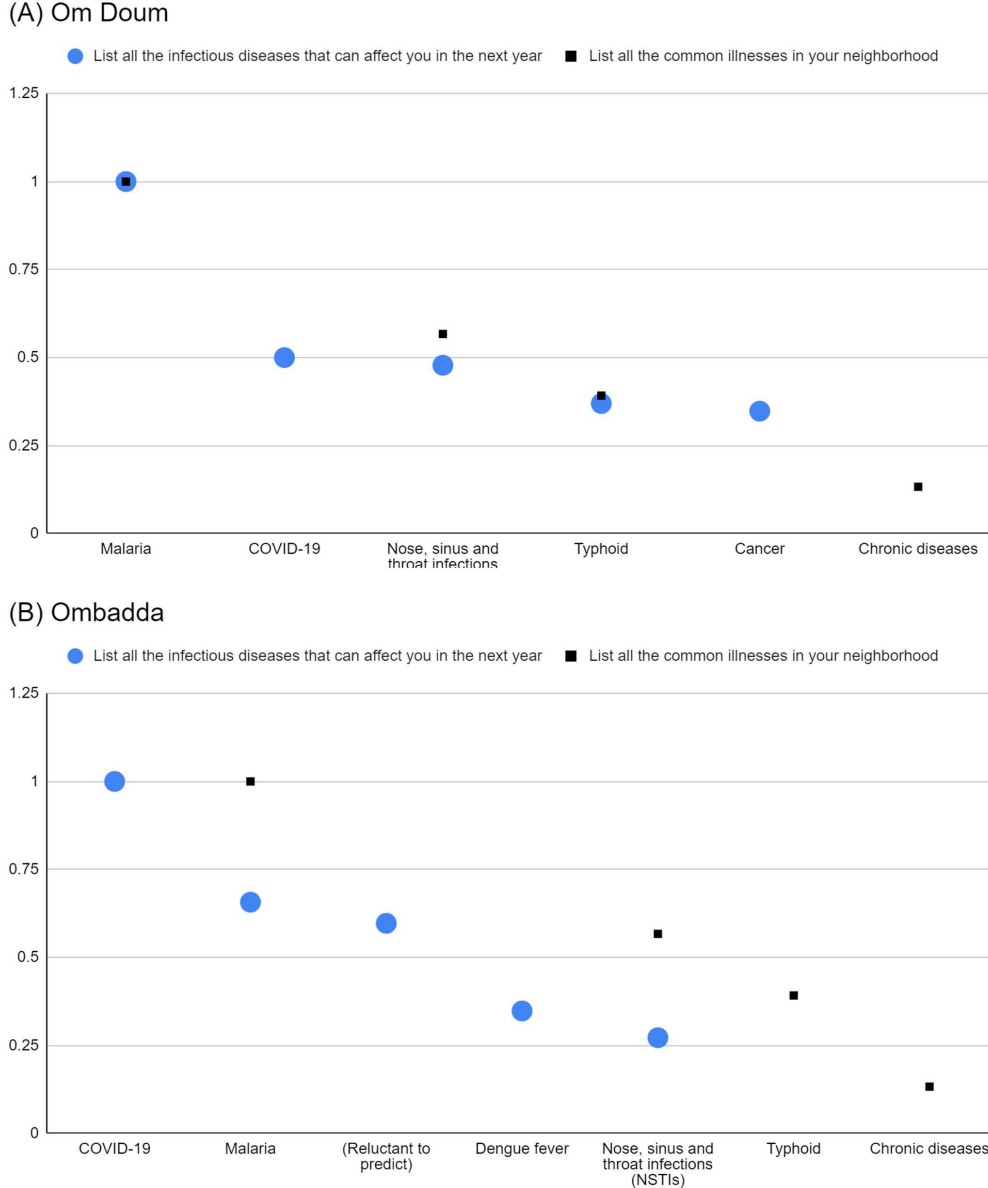

(A) Om Doum

(B) Ombadda

**Fig 5. Normalised salience indices of reported common illnesses (prompt 1) and predicted infections that could affect respondents (prompt 7) in Om Doum (A) and Ombadda (B).**

on its own (57.1 deaths per 100,000 population). This pattern mirrors findings from urban neighbourhoods in Ghana, where infectious diseases like malaria remain prevalent alongside an increase in non-communicable diseases [41]. In a Kenyan wetland community, respondents named malaria, typhoid, and diarrheal diseases in open-ended survey questions, along with joint pains, fluorosis, and other non-infectious conditions [42]. Similarly, during a typhoid fever outbreak in Malawi, illnesses highly rated for severity included typhoid fever, dysentery, and malaria, alongside non-infectious diseases like stroke, asthma, and heart disease [20]. Our findings indicate that respondents are cognizant of the complex landscape of disease distribution within their communities, offering opportunities for integrated rather than disease-specific risk communication interventions.

We also identified that both communities recognised recent local outbreaks. The government had responded to malaria, cholera and COVID-19 outbreaks in Khartoum during the three years before the study [43–45]. Our findings are similar to those of studies that prompted respondents to recall a specific outbreak event. For example, a quantitative study in a malaria-endemic district in Tanzania reported that almost two-thirds of respondents recalled a malaria epidemic that had occurred a year earlier [46]. Another quantitative study in Uganda reported that almost half of respondents were aware of an Ebola outbreak in the neighbouring Democratic Republic of Congo, with awareness significantly higher in districts at higher risk of Ebola importation [47]. Also, a narrative review of community-based surveillance systems in fragile and conflict-affected countries, many of which are epidemic-vulnerable, concluded that community members could successfully signal diseases with salient symptoms [25]. Our findings indicate that our study communities demonstrate a potential for community-powered epidemic surveillance to improve timely outbreak detection and response. However, with regards to COVID-19 as a novel disease, our study findings contrast with a study using focus group discussions reporting that denial of COVID-19 was still prevalent in similar communities in Khartoum in 2021, a year before our study [26]. Similarly, a study in Liberia using focus group discussions reported ongoing denial of Ebola in three Liberian communities four years after the peak of the 2014 epidemic [48].These findings suggest that using different research methods to engage people may generate different results related to people's recognition or denial of outbreaks in their communities.

Our results also show that respondents in both locations evaluate the perceived likelihood of contracting an illness and the perceived severity of that illness independently, with this distinction being more pronounced in Ombadda. They perceive a lower likelihood of contracting diseases they recognise as more severe. Conversely, they believe they are more likely to contract less severe illnesses. This phenomenon, known as 'unrealistic optimism' and described as a cognitive bias, involves a lower perceived likelihood of severe diseases and is frequently observed in the general population across various cultures [49,50]. This finding was also reported by a systematic review of public epidemic risk perceptions in epidemic-vulnerable countries [17]. Evidence suggests that optimistic bias can be addressed through risk communication by encouraging individuals to compare their risk with close peers from their own context, rather than with abstract "average persons" [51], and by using narrative risk communication that includes personal stories from the social context of the target audience [52]. Our study does not provide sufficient nuanced information to posit an explanation for the nuanced disease- or site-specific patterns or their consequences or implications for research or practice.

Our findings suggest that local disease manifestations significantly influenced the perceived likelihood and severity of infections, with this influence being consistent across diseases in Om Doum but inconsistent in Ombadda. This pattern aligns with previous observations in Sudan, such as a 2021 study where respondents cited local COVID-19 case numbers and deaths to explain their worry levels [26]. However, local epidemiology alone does not fully explain perceived risk, and its influence varies across diseases and populations. For example, a study conducted during a typhoid fever outbreak in Malawi found that the high number of local fatalities partly explained the high perceived severity [20]. Similarly, during the 2014 Ebola outbreak in Sierra Leone, perceived risk decreased over time, in line with the actual risk of transmission, due to social learning, improved protective behaviours, and better understanding through trusted community leaders [53]. These findings imply that while highlighting local disease burdens in risk communication is beneficial, its effectiveness in influencing protective behaviours varies over time and across diseases and communities.

Our findings suggest that Om Doum generally showed more consistency between respondents in interpretations of concepts such as "outbreaks" and "common illnesses", and on the salience of diseases across lists. We argue that this may be due to the higher social connectedness within Om Doum, compared to Ombadda, leading to a higher degree of shared understanding and perceptions of specific diseases. Our interpretation is also consistent with our other study finding that communication about epidemic-prone diseases in Om Doum occurred mainly within the community, while in Ombadda, it extended to people outside its confines. Other studies have previously reported on the influence of social network structures and degrees of social connectedness on various health outcomes, including preventive behaviours,

in low- and middle-income countries [54]. These findings indicate the importance of contextualising risk communication messages and channels for specific communities and diseases.

## Limitations

Our study has some limitations. We assumed that eligible residents in Om Doum and Ombadda are homogenous groups, whereas previous studies have shown that infectious disease risk perceptions can vary across cultural, linguistic, ethnic or religious groups [17,55]. Our findings suggest that respondents in each site were reasonably homogenous, especially in Om Doum, and while this served our analysis, it may also be due to partially recruiting participants from social gatherings. We could not compare responses by age, gender, and occupation within sites, as this would have required at least 20 respondents per population subgroup [19,37]. Also, we may have introduced bias during data cleaning, and categorisation of responses under synonyms and similar concepts, and the local meanings and ways of recognising diseases are likely to differ from some of the biomedical categories we assigned them to. To minimise this bias, we relied on study team members who are community members to lead the initial analysis by guiding the interpretation of the respondents' words during categorisation. Freelisting responses are lists of individual words or phrases and lack the depth of responses obtained from semi-structured interviews or focus group discussions. We audio-recorded interviews to obtain insight into participants' understandings of the question and their responses. We have also used the freelisting findings to guide further data collection on peoples' experiences and perceptions of epidemics in both sites, using individual interviews and focus group discussions in the study communities, which will be reported separately. As freelisting relies on participant recall, terms that are more recent or familiar may be relatively more salient – for example, during data collection, there was a seasonal increase in malaria cases in both study communities, which may have resulted in an overestimation of the salience of malaria. Finally, given our study design, findings cannot be generalised beyond the study sites; however, we provided a detailed description of the study sites to facilitate transferability of some of the findings to contexts with similar characteristics.

## Conclusions

Our findings highlight the shared perceptions of the study communities in Khartoum regarding epidemic-prone diseases, particularly malaria, typhoid, and COVID-19. Despite site-specific differences, these diseases consistently emerged as significant concerns, reflecting their epidemiological burden and perceived impact. Local disease patterns shaped risk perceptions, though interpretations varied. Our findings underscore the need for tailored risk communication interventions, informed by formative research, that address shared concerns through integrated multi-disease messaging, reflection of local disease burdens, contextualised peer-based risk comparisons, and the use of personal stories.

The study sites have undergone significant changes since the conflict escalation in April 2023, which likely influenced local disease dynamics and community priorities. These shifts emphasise the importance of continuously updating our understanding of local health concerns in rapidly changing contexts, particularly in areas experiencing displacement-induced demographic changes. While, in the short-term, research feasibility is complicated by mistrust, insecurity and ethical considerations, the findings of this study can serve as a pre-conflict baseline for researchers exploring disease perceptions by study populations in the future.

Freelisting demonstrated its utility as a rapid method for identifying risk perceptions of epidemic-prone diseases, yielding actionable insights into concerns and intervention options. However, while it is a practical approach, we recommend its use alongside complementary methods, such as focus groups or interviews, which can provide a richer context and further validate findings. By integrating these findings and approaches, actors involved in epidemic preparedness and response can better align interventions with community perceptions and needs, enhancing both the relevance and impact of their programs.

## Supporting information

**S1 Text.  This file is the data collection instrument.**
(DOCX)

**S2 Text.  This file contains a comprehensive list of all terms mentioned by participants at each site, and their grouping into broader terms as part of data curation prior to analysis.**
(DOCX)

**S3 Text.  This file contains seven figures - 1 per freelist - comparing the normalised salience indices of the most salient terms in Om Doum and Ombadda.**
(DOCX)

## Author contributions

**Conceptualization:** Nada Abdelmagid, Bayard Roberts.

**Data curation:** Nada Abdelmagid, Omama Abdalla, Abdallah Yagoub, Abeer Taha, Altayeb Hyder, Awatif Yahia, Bashar Hassan, Marwa Ali, Mohammed Abdeen, Mustafa Adam, Nadeen Kamal, Nader Ezeideen, Osama Altib, Rahma MohamedSalih, Salma Alnour, Sana Koko, Shama Abdelatif, Waleed Yahia, Yousif Abdelhade.

**Formal analysis:** Nada Abdelmagid.

**Investigation:** Omama Abdalla, Abdallah Yagoub, Abeer Taha, Altayeb Hyder, Awatif Yahia, Bashar Hassan, Marwa Ali, Mohammed Abdeen, Mustafa Adam, Nadeen Kamal, Nader Ezeideen, Osama Altib, Rahma MohamedSalih, Salma Alnour, Sana Koko, Shama Abdelatif, Waleed Yahia, Yousif Abdelhade.

**Methodology:** Nada Abdelmagid, Omama Abdalla, Abdallah Yagoub, Abeer Taha, Altayeb Hyder, Awatif Yahia, Bashar Hassan, Marwa Ali, Mohammed Abdeen, Mustafa Adam, Nadeen Kamal, Nader Ezeideen, Osama Altib, Rahma MohamedSalih, Salma Alnour, Sana Koko, Shama Abdelatif, Waleed Yahia, Yousif Abdelhade.

**Project administration:** Nada Abdelmagid, Omama Abdalla.

**Supervision:** Nada Abdelmagid, Bayard Roberts.

**Visualization:** Nada Abdelmagid.

**Writing – original draft:** Nada Abdelmagid.

**Writing – review & editing:** Omama Abdalla, Jennifer Palmer, Michelle Lokot, Bayard Roberts.

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
