## [Decision Letter · Decision Letter 0]

PGPH-D-25-00084

Salience and perceptions of epidemic-prone diseases in two communities: findings from freelisting interviews in Khartoum State, Sudan

Dear Dr. Nada Abdelmagid,

Thank you for submitting your manuscript to PLOS Global Public Health. After careful consideration, we feel that it has merit but does not fully meet PLOS Global Public Health’s publication criteria as it currently stands. Therefore, we invite you to submit a revised version of the manuscript that addresses the points raised during the review process.

We look forward to receiving your revised manuscript.

Kind regards,

Muhammad Asaduzzaman, MD MPH MPhil

Academic Editor

Journal Requirements:

Additional Editor Comments (if provided):

Reviewers' comments:

Reviewer's Responses to Questions

**Comments to the Author**

1. Does this manuscript meet PLOS Global Public Health’s publication criteria ? Is the manuscript technically sound, and do the data support the conclusions? The manuscript must describe methodologically and ethically rigorous research with conclusions that are appropriately drawn based on the data presented.

Reviewer #1: Yes

Reviewer #2: Yes

Reviewer #3: Yes

Reviewer #4: Partly

2. Has the statistical analysis been performed appropriately and rigorously?

Reviewer #1: Yes

Reviewer #2: Yes

Reviewer #3: Yes

Reviewer #4: Yes

3. Have the authors made all data underlying the findings in their manuscript fully available (please refer to the Data Availability Statement at the start of the manuscript PDF file)?

Reviewer #1: Yes

Reviewer #2: Yes

Reviewer #3: Yes

Reviewer #4: No

4. Is the manuscript presented in an intelligible fashion and written in standard English?

Reviewer #1: Yes

Reviewer #2: Yes

Reviewer #3: Yes

Reviewer #4: Yes

5. Review Comments to the Author

Reviewer #1: 1. Is the manuscript technically sound, and does the data support the conclusions?

It presents a technically sound study. The use of freelisting is appropriate for the research objective, which is to understand local perceptions of epidemic-prone diseases. The findings are logically presented and supported by descriptive data analysis. The authors are careful in framing conclusions, particularly in the discussion, where they acknowledge the limitations of freelisting and highlight the need for complementary methods.

2. Has the statistical analysis been performed appropriately and rigorously?

The statistical approach, specifically the use of Smith’s salience index and a relative salience index is appropriate for freelisting data. The authors explain the analytical process clearly.

3. Have the authors made all data underlying the findings in their manuscript fully available?

All relevant data are included in the manuscript and its supporting information files. The inclusion of the S1–S3 Appendices supports this claim and provides sufficient detail for replication or further inquiry.

4. Is the manuscript presented in an intelligible fashion and written in standard English?

The manuscript is generally well-written and intelligible. The use of language is professional and meets the standards of academic writing. However, some sections, particularly the comparative analysis across prompts (tables 2–5, and figures 1–5), are dense and may benefit from a concise presentation or reorganization.

Additional comments

This manuscript contributes significantly to the literature on community perceptions of epidemic risk and demonstrates methodological rigor. With minor improvements in clarity and further elaboration on the following observations, it will be well-positioned for publication.

1. The purposive sampling strategy is acceptable, but further detail is needed on how the diversity of participants was ensured (e.g., with respect to age, gender, and occupation). Recruiting participants from social gatherings may introduce homogeneity, which should be acknowledged as a limitation.

2. While the limitations of freelisting are discussed in the discussion section, a brief mention in the methods section would provide better context for readers unfamiliar with the technique’s constraints, particularly its reliance on recall and limited depth.

3. Presentation of Tables:

o Table 2 is informative but overcrowded. Consider splitting it for better readability.

o The category ‘reluctant to predict’ in Table 4 is socio-culturally significant. A brief explanatory footnote to contextualize this response would enhance interpretation.

o Including brief footnotes or definitions for local or technical terms (e.g., NSTIs) would help readers unfamiliar with the setting.

4. Interpretation of site differences between Om Doum and Ombadda largely attributes variations to social cohesion. This claim would be more persuasive with additional evidence or references to known social structure differences between the sites.

5. The discussion around unrealistic optimism is compelling and could be further strengthened by suggesting how such cognitive biases might be addressed through targeted community engagement or epidemic preparedness efforts.

6. The mention of the post-April 2023 conflict is appropriate but underdeveloped. Expanding on how this shift could impact future research feasibility, disease salience, or community risk perception would increase the relevance and timeliness of the study.

Reviewer #2: General Comments:

This manuscript presents an important and original contribution to understanding local perceptions of epidemic-prone diseases in Sudan. The use of freelisting interviews in both urban and rural settings is appropriate for the research question and provides valuable insights for tailoring health interventions and risk communication strategies in local contexts.

The study is technically sound, methodologically appropriate, and generally well-written. The authors have correctly applied a recognized analytical approach (Smith’s S Salience Index) to identify diseases most salient to community members. The conclusions are well supported by the data and are appropriately cautious given the qualitative nature of the study.

However, there are some minor areas for improvement. The authors briefly acknowledge study limitations but could benefit from expanding this section to discuss issues such as context specificity, potential recall bias and limited generalizability beyond the study settings

Reviewer #3: Recommendation for Manuscript PLOS - # PGPH-D-25-0084

The authors of the manuscript lay out a clear rational and justification for carrying out the study – Salience and Perceptions of Epidemic-prone Diseases in Two Communities in Khartoum State, Sudan. The study was an exploratory endeavor to assess the feasibility of a qualitative technique; freelisting interviews as a tool for measuring salience and capturing risk perceptions of epidemic-prone diseases in conflict environments.

The rational provided is that understanding local perceptions of epidemic-prone diseases would assist in designing of effective, participatory and culturally appropriate epidemic responses; improve risk communication and foster trust in professional epidemic responses among the community.

The authors designed a robust study, collected appropriate data using freelisting interviews and conducted thorough analysis of the data, utilizing the correct analysis procedures for measuring both Smith’s and Relative salience across items and participants – as well as appropriate qualitative analysis procedures for capturing participants perceptions on the relevant concepts of interest.

However, while the limitations of the study are described and results are clear, the interpretation falls short of fully explaining the caveat expressed in the main finding – “… epidemic-prone diseases are perceived as significant concerns, [BUT] not sufficient for all diseases…” . This statement requires unpacking to make the message being conveyed clearer, especially because the authors posit that the freelisting tool/technique “…requires complementary methods to explore nuanced patterns…”.

I invite the authors to consider minor revisions of the interpretation to ensure readers understand that freelisting interviews lack depth even as they provide broad local knowledge of a domain, and therefore, must be used in a mixed-method design plan in order to gain results that would guide intervention planning. The tool lends itself well to a formative phase of a study because it allows for a broad understanding of a topic like epidemic-prone diseases. However, to enhance the utility of the results from freelisting interviews, especially the salience of items; they should be used to develop structured survey instruments or key-informant and focus group guides; – techniques which provide robust results with depth, which would in turn better assist in designing appropriate context-sensitive interventions with targeted messages in response to epidemics in conflict settings.

Recommendation: Accept with minor revisions

Reviewer #4: Summary and overall impression

The manuscript by Abdelmagid et al. reports salience and perceptions of epidemic-prone diseases among members of two urban communities in Sudan, a relevant topic given the existing lack of published literature. The authors used clear language and sound methods to obtain and report their findings. A major strength of the study is the participatory approach used in the design of prompts and the data collection and cleaning. Nevertheless, some sections of the manuscript are not entirely clear to the reader and need more detailed explanations, and some interpretative parts of the results section would be better suited if moved to the discussion section. In addition, the discussion and conclusion would benefit from focussing on the public health implications of the results and explaining the authors' recommendations in more detail in this sense, including specific examples. Therefore, I recommend to accept this manuscript for publication after major revision.

Discussion of specific areas for improvement

Major issues

1. Interpretive parts in results section

a. In the description of Table 3, there are some interpretive parts (“…, indicating a general recognition of recent epidemic events among participants.” and “…although health authorities had reported an ongoing monkeypox outbreak based on two confirmed cases in Khartoum at the time of data collection..”) which require references and might be better suited in the discussion section. Additionally, I recommend to use the term “Mpox” rather than “monkeypox”, as the former is currently commonly agreed upon.

b. Similarly, the description of Table 4 contains interpretive parts which should be moved to the discussion: “…possibly reflecting cultural and faith-related perspectives on predictions and destiny.” The provision of quotations of study participants who were reluctant to make predictions is enhancing comprehensibility. Yet, harmonizing their formatting and better embedding them in the body text through the provision of an introductory sentence would further enhance readability.

c. Comparison of prompt 1 and prompt 3: Similar as above the following interpretive parts would be better suited in the discussion: “This suggests that these diseases might be perceived as more noteworthy than their actual prevalence within the neighbourhood would suggest.” and “This finding raises the possibility that discussions in the population may prioritise ‘severe’ illnesses, even if they are not perceived as highly common in the neighbourhood.”

d. Comparison of prompt 1 and prompt 4: Similar as above, I would recommend moving the interpretive parts “These findings suggest diverse interpretations of the concepts of ‘outbreaks’ and ‘common illnesses’..” and “The findings above suggest that the study populations recognise epidemics through the observed behaviour of the disease within the community,…” and “This may be influenced by the timing of data collection and phrasing of the prompts, as COVID-19 cases had reduced locally at the time of the study. Still, it had received significant public attention in the three years preceding data collection.” to the discussion section whereby the latter also requires references.

e. Comparison of prompt 1, prompt 5 and prompt 7: Similar as above, I would recommend moving the interpretive parts “These findings suggest a potentially strong influence of local disease epidemiology on the perceived probability of infection and disease severity among Om Doum respondents” and “These findings suggest that local disease epidemiology was not consistently influential on Ombadda participant when …” and “These findings indicate that, even with reduced or absent local cases, participants did not discount..” to the discussion section.

2. The discussion is generally well structured and compares findings with relevant literature from other countries. However, it would further benefit from the following suggestions:

a. A comparison of findings regarding common diseases with local surveillance data would be interesting. This would allow the reader to understand how well community perceptions align with epidemiological findings.

b. Regarding the section on outbreaks, it would be interesting to add some surveillance information about how many outbreaks occurred in the time before the study, what diseases were most frequently causing outbreaks and what were the most severe ones in terms of disease burden. This would allow for a comparison of epidemiological data with the salience indices found for outbreaks in the study.

c. The authors mention implications of their findings for public health policy. However, more details on specific measures or actions that their results imply would add relevance to the discussion. This could include a discussion of how current measures taken in Khartoum in terms of risk communication, epidemic preparedness and response could be altered and improved based on the findings of the study.

3. The limitation section transparently describes the issues that might have caused bias in the study. It would be interesting to further elaborate how the selected study population, i.e. urban communities in Khartoum State, might influence findings and if the latter might differ among e.g. rural communities, and what implications that has for the generalisability and applicability of findings.

4. The conclusions section highlights main findings but its relevance could – similarly to the discussion section – be enhanced by more clearly stating public health policy implications of the study’s findings. Additionally, it is not completely clear how the authors reached the conclusion that “particularly environmental measures for diseases like malaria and typhoid” are needed as this is mentioned here for the first time. A more detailed elaboration of how the findings led to this conclusion in the discussion section could back up this claim.

Minor issues

1. Clarifications

a. The comprehensibility of the manuscript and findings would be enhanced by adding a definition of epidemic-prone diseases as understood by the authors.

b. Prompts:

i. Prompt 5 and prompt 7 seem very similar, although it is understandable that authors aimed for a difference in meaning. It would be beneficial to add a short explanation of what was meant exactly by each prompt and how data collectors ensured that participants understood the prompts correctly.

c. Study setting:

i. The participatory approach involving local volunteers is a major strength. To enhance comprehensibility, a short description of the process of recruiting those volunteers and their characteristics would be beneficial.

ii. While study sites are well described, this part might benefit from a short explanation about why those sites were chosen. Although potentially difficult, the descriptions would be further justified by adding references.

d. Study participants:

i. For a better understanding and reproducibility of findings, the term “eligible” could be described in more detail by stating the applied inclusion and exclusion criteria.

e. Analysis:

i. The calculated measures are well described and follow best practice. However, the terminology used in the method section does not fit what is reported in the results, e.g. relative salience indices are stated in methods whereas normalised salience indices are reported in the results. Using a common terminology throughout the manuscript would enhance comprehensibility.

ii. The involvement of community members in data cleaning is a strength of the study as it reduces bias. However, it remains unclear why this was only performed for prompt 4 and 6 and not for all prompts.

iii. The stepwise procedure of data analysis is generally a suitable method in qualitative research. However, the description in the methods section would benefit from adding more details on how the preliminary analysis the authors conducted informed further research.

iv. Additionally, adding more details on the data cleaning and translation procedure would be helpful for better understanding. For example, how many researchers were involved? Was cross-checking performed? At what point during the analysis were terms translated to English? Was more than one translation produced and results reviewed against each other?

f. Data collection:

i. The description of the exact period of data collection, number of interviews conducted and length of interviews would be more suitable at the beginning of the results section than in the methods section. This allows for a comparison of the “target” and the “actual”.

2. Figures and tables

a. Table 1: Adding the total number of interviews per study site would enhance comprehensibility.

b. Stating the prompts in the table captions is enhancing readability. To standardize the format, prompts in captions of Table 3 and Table 5 should also be put in quotes like in other table captions.

c. Table 5: The description of results in Table 5 are not completely clear. Understanding would be improved by explaining the remark about the extended family in Om Doum, as this is not completely evident from the table. Additionally, comprehensibility could be improved by adding more details on why the listed terms indicate that that “communication occurs within the confines of the neighbourhood and its population” in Om Doum and why this is different to Ombadda where “communication about outbreaks transpires within the neighbourhood and the broader district..”.

3. Additional missing references

a. The following part of the discussion would require a reference: “The government had declared malaria and COVID-19 outbreaks in Khartoum during the three years before the study.”

6. PLOS authors have the option to publish the peer review history of their article (what does this mean? ). If published, this will include your full peer review and any attached files.

**Do you want your identity to be public for this peer review?** For information about this choice, including consent withdrawal, please see our Privacy Policy .

Reviewer #1: **Yes**

Reviewer #2: **Yes**

Reviewer #3: No

Reviewer #4: No

---

## [Decision Letter · Decision Letter 1]

Salience and perceptions of epidemic-prone diseases in two communities: findings from freelisting interviews in Khartoum State, Sudan

PGPH-D-25-00084R1

Dear Nada Abdelmagid,

We are pleased to inform you that your manuscript 'Salience and perceptions of epidemic-prone diseases in two communities: findings from freelisting interviews in Khartoum State, Sudan' has been provisionally accepted for publication in PLOS Global Public Health.

Best regards,

Muhammad Asaduzzaman, MD MPH MPhil

Academic Editor

Reviewer Comments (if any, and for reference):

Reviewer's Responses to Questions

**Comments to the Author**

1. If the authors have adequately addressed your comments raised in a previous round of review and you feel that this manuscript is now acceptable for publication, you may indicate that here to bypass the “Comments to the Author” section, enter your conflict of interest statement in the “Confidential to Editor” section, and submit your "Accept" recommendation.

Reviewer #1: All comments have been addressed

Reviewer #4: All comments have been addressed

2. Does this manuscript meet PLOS Global Public Health’s publication criteria ? Is the manuscript technically sound, and do the data support the conclusions? The manuscript must describe methodologically and ethically rigorous research with conclusions that are appropriately drawn based on the data presented.

Reviewer #1: Yes

Reviewer #4: Yes

3. Has the statistical analysis been performed appropriately and rigorously?

Reviewer #1: Yes

Reviewer #4: Yes

4. Have the authors made all data underlying the findings in their manuscript fully available (please refer to the Data Availability Statement at the start of the manuscript PDF file)?

Reviewer #1: Yes

Reviewer #4: Yes

5. Is the manuscript presented in an intelligible fashion and written in standard English?

Reviewer #1: Yes

Reviewer #4: Yes

6. Review Comments to the Author

Reviewer #1: Review Comments to the Author

1. Sampling strategy and participant diversity

My comment about participant diversity acknowledges potential homogeneity due to recruitment at social events. The revised manuscript's “Study participants” section explains the purposive sampling approach and notes that many participants were recruited from events like health education campaigns, coffee gatherings, and public places (e.g., barbershops), which may lead to some homogeneity. The section also provides demographic details (gender, age, occupation), strengthening the diversity claim.

2. Limitations of freelisting should appear in the Methods

My earlier comment on freelisting limitations is addressed. The “Study design” section now briefly notes reliance on recall and limited depth as limitations of freelisting, aligning with your suggestion.

3. Table presentation issues

• Table 2 overcrowded? Still dense but visually improved with clearer headers. They haven’t split the table, but they’ve better formatted it. So, it is partially addressed.

• ‘Reluctant to predict’ in Table 4 is addressed. It now includes a clear cultural explanation and two direct quotes to contextualize this salient response.

• Definitions for local/technical terms like NSTIs are addressed. NSTIs are now clarified as “nose, sinus, and throat infections,” which helps non-local readers.

4. Interpretation of site differences (Om Doum vs. Ombadda)

My comment on claims regarding social cohesion needing evidence is partly addressed. The authors expanded on the social composition of both sites, suggesting differences in cohesion (e.g., Om Doum’s close-knit, multi-generational households vs. Ombadda’s urban diversity). However, they still heavily rely on inference instead of complex data or external references.

5. Unrealistic optimism and cognitive bias

My comment about expanding the discussion on this and suggesting interventions is acknowledged. The authors not only clarify the phenomenon (especially the salience inversion between probability and severity) but also recommend enhancing risk communication through tailored community engagement.

6. Impact of the April 2023 conflict

My comment to expand the discussion of its implications for risk perception and future research is addressed. In the Discussion, the authors elaborate on how ongoing conflict could influence community disease salience and the feasibility of conducting similar studies in the future.

Reviewer #4: Thank you for considering my comments. They have been satisfactorily addressed and I recommend that the manuscript be accepted for publication.

7. PLOS authors have the option to publish the peer review history of their article (what does this mean? ). If published, this will include your full peer review and any attached files.

**Do you want your identity to be public for this peer review?** For information about this choice, including consent withdrawal, please see our Privacy Policy .

Reviewer #1: **Yes: ** Yitaferu, Tadele B

Reviewer #4: No
